# Examining the Impact of the COVID-19 Pandemic on Suicide-Attempt Survivors

**DOI:** 10.3390/ijerph22071072

**Published:** 2025-07-04

**Authors:** Martina Fruhbauerova, Julie Cerel, Athena Kheibari, Alice Edwards, Jessica Stohlmann-Rainey, Dese’Rae Stage

**Affiliations:** 1Department of Psychology, University of Kentucky, Kastle Hall, 106B, 503 Library Dr, Lexington, KY 40508, USA; 2College of Social Work, University of Kentucky, 619 Patterson Office Tower, Lexington, KY 40508, USA; julie.cerel@uky.edu (J.C.);; 3School of Social Work, Wayne State University, 5447 Woodward Ave, Detroit, MI 48202, USA; athena.kheibari@wayne.edu; 4Solutions by Jess, 3505 N Elizabeth St., Denver, CO 80205, USA; 5Live Through This Productions, LLC, 2310 Mercer St., Philadelphia, PA 19125, USA

**Keywords:** COVID-19, pandemic, lived experience, suicide-attempt survivor, resiliency

## Abstract

Despite initial concerns about the severe negative impact of COVID-19 on individuals with a history of mental health problems and suicide attempts, its effects remain unclear. This study examined the pandemic’s impact on individuals with and without lived experience of suicide attempts. An online nationwide sample of 1351 adults from the United States completed questionnaires from 26 May to 25 June 2021. A history of suicide attempt(s) (n = 159; 12%) was associated with significantly higher odds of utilizing mental health services, hospitalization for psychiatric reasons, and contacting hotlines. This history predicted worse outcomes in functioning, optimism, despair, and impairment. Notably, 57.6% of these individuals believed surviving a suicide attempt made them more resilient, while 21.9% expressed uncertainty about its impact on their resilience. In sum, participants with a history of suicide attempt(s) reported more depressive symptoms, worse daily functioning, more despair, less optimism, and greater service utilization during the pandemic, yet many also cited increased resilience due to their suicide history.

## 1. Introduction

Early in the COVID-19 pandemic, there was public apprehension that it would lead to adverse psychiatric effects in the general population as well as among individuals with preexisting mental illnesses [1]. It was hypothesized that the pandemic would be associated with increased stress, anxiety, and suicide deaths [1,2]. A report from early in the pandemic estimated that an additional 75,000 “deaths of despair” in the U.S. could result from economic and social fallout from the virus [3]. Indeed, a review of studies during the first year of the pandemic found that anxiety, depression, and distress increased worldwide in the early months as a result of the pandemic [4,5].

However, despite these widespread concerns, suicide rates, life satisfaction, and loneliness remained largely stable throughout the first year of the pandemic [4]. Indeed, for the first time in over a decade, suicide was not in the top ten causes of death in the U.S. in 2020. In fact, reports show that suicide deaths decreased in 2020 in the U.S. [6] and in most countries around the world [7]. However, it is important to note that minoritized racial and ethnic groups experienced increases in suicide deaths [8]. The COVID-19 outbreak, which resulted in business closures, increased rates of unemployment/decreased income, housing concerns, and mandated stay-at-home orders, brought the issue of suicide risk to the forefront of public concern, with news outlets discussing mental health, suicide prevention strategies, and the importance of reaching out to loved ones. Non-COVID-19 hospitalizations dropped below typical levels [9]—in part due to overburdened units and restrictions of numbers due to distancing with COVID-19 patients.

Although suicide-death rates did not increase in 2020, crisis- and suicide-hotline usage did increase in the U.S. and other countries [10]. Emergency department visits for suicidal ideation and attempts also made up a larger percentage of visits than in comparable time periods the previous year, as emergency department visits for other reasons dropped substantially [11]. It is possible that help-seeking and the utilization of crisis services might have protected individuals from dying by suicide. Alternatively, a recent qualitative study of individuals experiencing suicidal ideation during the pandemic found that, for some people, the pandemic created barriers to carrying out their suicide plans, such as the inability to obtain lethal means [12].

Given that the majority of suicide attempt decedents and survivors have a diagnosable mental health condition [13,14,15], it is reasonable to suspect that the effects of the pandemic might have been especially distressing for them. However, there is reason to believe that, for some people, the experience with isolation, and mental health challenges prior to the pandemic might have led to more resilience in the face of COVID-19 than for people who had not had these experiences. For example, a popular press article authored by a person with bipolar disorder written early in the pandemic states, “This is not a drill, but for those of us who struggle with bipolar disorder like myself, or depression, or anxiety, this is not our first rodeo. We’re surprisingly copacetic. We’ve had emotional distress time after time, so we are more resilient than the normies a.k.a. people without mental illness. In other words, we’ve been training for this our whole lives” [16].

However, the experiences of those who had previously made suicide attempts have not yet been examined. The need for up-to-date information on the impacts of COVID-19 on suicide-attempt survivors to inform health system responses is imperative.

***Current Study***. The current study sought to investigate the effects of the pandemic on individuals with lived experience of suicide attempts compared to those without lived experience. Given the limited empirical data on this topic, no a priori hypotheses were made about these outcomes. Additionally, the present study also aimed to examine perceived resilience during the pandemic and assess disparities in the utilization of mental health resources between the two groups. Based on anecdotal evidence suggesting that prior experiences of isolation and mental health challenges might foster resilience, we hypothesized that individuals with a history of suicide attempts would perceive themselves as more resilient during the pandemic than those without such a history. Further, considering the documented increase in emergency room visits for suicidal ideation during the pandemic, we hypothesized that individuals with a history of suicide attempts would be more likely to seek support from mental health professionals, utilize psychiatric hospital services, and contact crisis hotlines compared to those without such a history.

## 2. Materials and Methods

This cross-sectional study utilized CloudResearch, an online crowdsourcing platform that connects researchers with participants through Amazon Mechanical Turk (MTurk) [17]. MTurk is an online marketplace operated by Amazon where individuals, known as “workers,” complete various tasks—including surveys and psychological studies—in exchange for monetary compensation. CloudResearch serves as an intermediary, improving the recruitment process by allowing researchers to screen participants based on specific demographic or behavioral criteria, ensuring higher-quality data. Both MTurk and CloudResearch have become widely used in survey-based research due to their cost-effectiveness, rapid data collection, and access to a diverse participant pool. Previous studies have demonstrated the reliability of these platforms for recruiting participants and collecting high-quality psychological and behavioral research data [18,19,20]. MTurk’s workers are demographically very similar to respondents on other survey platforms [21] and may be representative of a larger population if no specific qualifications are required of them [22]. Participants were contacted via MTurk, and only individuals on CloudResearch’s Approved List—pre-vetted participants who have demonstrated high reliability and data quality—were invited to participate. Workers access surveys through the MTurk platform by accepting posted tasks, which include a description of the study, compensation details, and a link to the survey hosted on Qualtrics. A total of 1467 participants accessed the survey and provided complete and valid responses. Random sampling was ensured by setting no specific qualifications beyond inclusion on the Approved List and leveraging the randomization tools available through CloudResearch. Efforts to represent participants from across the 50 states were made by utilizing the geographical and demographic diversity inherent to MTurk’s worker pool, which has been shown to include individuals from all 50 states. The Qualtrics XM software platform [23] was used to create the online survey and generate an anonymous survey link. Study data were collected from 26 May to 25 June 2021, and all participants received USD 1.50 for completing the main survey. A subset of participants who endorsed a history of suicide attempts was eligible for follow-up questions, for which they received an additional USD 3.25. The entire survey took approximately 25 minutes to complete. Informed consent was collected prior to the collection of any data. All procedures were reviewed and approved by the University of Kentucky Institutional Review Board (IRB).

### 2.1. Participants

A nationwide sample of 1467 adults from 50 U.S. states was collected. Inclusion criteria for participation included (a) being a U.S. resident, (b) at least 18 years of age, (c) having at least a 90% approval rating as an MTurk worker, and (d) completion of at least 500 previously approved MTurk tasks to ensure high data quality. A total of 116 participants did not indicate whether they had ever made a suicide attempt in their lifetime and were excluded. The final sample included 1351 participants. All authors confirm that they have obtained appropriate consent for the publication of this manuscript. All data involving participants in this study have been anonymized to protect confidentiality. Participants provided informed consent for their data to be published in aggregate form.

### 2.2. Measures

Through sets of closed and open-ended self-report questions, this study investigated the effects of the COVID-19 pandemic on those with and without a history of suicide attempt(s) and suicide-attempt survivors’ perceptions of their resiliency. Pandemic effects were operationalized using four indicators: (1) visits to a mental health professional, (2) psychiatric hospitalizations, (3) contact with hotlines, and (4) a set of variables that we refer to as “pandemic impact,” described below.

### 2.3. Demographics

Participants provided demographic information, including age, race, ethnicity, education level, sexual orientation, and marital status.

### 2.4. Suicide

The lifetime history of a suicide attempt was assessed using a single-item measure. Participants were asked, ‘Have you ever attempted suicide (e.g., intentionally hurt yourself with at least some intent to die)?’ This question was adapted from the Self-Injurious Thoughts and Behaviors Interview (SITBI) Short Form [24]. Response options were binary (Yes/No). The primary purpose of this measure was to identify individuals with a history of suicide attempts and for inclusion in the follow-up questions of the study.

### 2.5. Resilience

Following a set of questions directly related to the ongoing pandemic, participants with a history of suicide attempt(s) who agreed to the follow-up questions were asked whether they perceived themselves as more or less resilient during the pandemic. The researchers (J.C. and A.K.) developed this question specifically for the purpose of this study. The single-item question asked participants: “Has surviving a suicide attempt made you more resilient?” Participants rated their responses on a five-point Likert scale ranging from 1 (“Definitely yes”) to 5 (“Definitely not”), with higher ratings indicating a perception of decreased resilience. This question aimed to capture participants’ subjective assessments of how their experiences with surviving a suicide attempt might have influenced their ability to cope with the unique challenges posed by the pandemic.

An adapted self-report version of the National Survey on Drug Use and Health (NSDUH) [25] was used to assess participants’ mental health service utilization. Specifically, the adapted items measured participants’ engagement in key areas, including (1) visits to mental health professionals, (2) psychiatric hospitalizations, and (3) contact with support services such as hotlines. Four items were used to evaluate outpatient treatment engagement, the use of psychiatric medications, hospitalizations for psychiatric reasons, and the utilization of crisis hotlines. These measures provided insight into the breadth of mental health support services accessed by participants.

#### 2.5.1. Mental Health Professional Visits

The participants were asked about their mental health professional visits during the pandemic, specifically whether they had seen a therapist or psychiatrist via telehealth or in person. Telehealth visits were assessed with questions such as, “Since March 2020, have you seen a therapist by telemedicine?” and “Since March 2020, have you seen a psychiatrist or someone who prescribes psychiatric medication by telemedicine?” Similarly, in-person visits were evaluated with questions such as, “Since March 2020, have you seen a psychiatrist in person?” or “Since March 2020, have you seen a therapist in person?” The response options were binary (Yes/No). These four variables—telehealth visits to a therapist or psychiatrist and in-person visits to a therapist or psychiatrist—were collapsed into a single composite variable, “mental health professional visits,” which captured any engagement with a therapist or psychiatrist through either modality. For participants who reported any visits, follow-up questions assessed the total number of visits made during the pandemic.

#### 2.5.2. Psychiatric Hospitalizations

The participants were asked to report any psychiatric hospitalizations they experienced during the pandemic with the question: “Since March 2020, have you been hospitalized in a psychiatric hospital?” If the participants responded affirmatively, they were prompted to provide the total number of hospitalizations. Responses to the initial question were recorded as binary (Yes/No), allowing for clear identification of hospitalization history during the specified timeframe.

#### 2.5.3. Contact with Hotlines

The participants were asked about their use of crisis and support hotlines since the onset of the pandemic in March 2020. Specifically, they were queried about contacting the National Suicide Prevention Lifeline (“Since March 2020, have you called the National Suicide Prevention Lifeline?”), the Crisis Text Line (“Since March 2020, have you texted the Crisis Text Line?”), other hotlines (“Since March 2020, have you used a hotline?”), and warmlines or peer-support lines (“Since March 2020, have you contacted a warmline or peer support line?”). For each question, the responses were recorded as binary (Yes/No). These individual variables were then collapsed into a single composite variable, labeled “contact with hotlines,” representing any engagement with crisis or support services. Participants who endorsed hotline contact were further asked to indicate their usage frequency.

#### 2.5.4. Pandemic Impact Variables

The final aspect of the effects of the COVID-19 pandemic was conceptualized through a set of variables referred to as “pandemic impact.” These variables encompassed scales assessing the overall impact of COVID-19 on daily functioning, levels of optimism amidst the pandemic, feelings of despair caused by the pandemic, and the extent of impairment experienced as a result of the pandemic.

4a. COVID-19 Impact on Functioning. The authors developed a scale to assess the overall impact of COVID-19 on daily living and functioning, specifically for this study. This scale was loosely inspired by the Impact of Event Scale-Revised (IES-R) [26] and consists of 10 items measuring the pandemic’s effects on various aspects of daily life (e.g., “Due to COVID-19, you live in constant fear.”). The participants rated each item on a six-point Likert scale ranging from 1 (“Not at all”) to 6 (“Completely”), resulting in possible total scores ranging from 10 to 60, with higher scores indicating a greater negative impact on functioning. The full scale is available in the Appendix A. In this study, the scale demonstrated acceptable internal consistency (Cronbach’s α = 0.67).

4b. Optimism About Overcoming Pandemic. Optimism toward the coronavirus/COVID-19 was assessed using items adapted from a GWI consumer research study on how individuals reacted to the COVID-19 situation [27]. For the present study, we adapted these questions to target optimism toward four sources: self, family/people you care about, local community, and country of residence. As such, the authors developed four questions that were worded identically to the GWI survey items except for whom the optimism was expressed (e.g., “How optimistic are you that your community will overcome the coronavirus/COVID-19 situation?”). The scale was rated on a Likert scale from 1 (“Not at all”) to 7 (“Completely optimistic”), resulting in possible total scores ranging from 4 to 28, with higher scores indicating greater optimism. In this study, the scale demonstrated good internal consistency (Cronbach’s α = 0.88).

4c. Feelings of Despair Due to Pandemic. The authors developed a measure of pandemic-related despair specifically for this study. This scale was loosely inspired by the Coronavirus Anxiety Scale (CAS) [28]. This measure assessed the frequency of hopelessness, helplessness, and despair related to the pandemic through five items created for this research (e.g., “I felt extremely hopeless about the future after thinking about the coronavirus in the last few weeks”). The participants rated each item on a four-point Likert scale ranging from 1 (“Not at all”) to 4 (“Nearly every day over the last 2 weeks”), resulting in possible total scores ranging from 4 to 20, with higher scores reflecting greater feelings of despair. The full scale is available in the Appendix A. In this study, the scale demonstrated good internal consistency (Cronbach’s α = 0.79).

4d. Impairment Due to Pandemic. Pandemic-related impairment was assessed by the Work and Social Adjustment Scale (WSAS) [29], a five-item scale designed to evaluate the impact of a person’s mental health difficulties on their ability to function in terms of work, home management, social leisure, private leisure, and personal or family relationships. This scale was rated on a Likert scale from 0 (“Not at all”) to 8 (“Very severely”), resulting in possible total scores ranging from 0 to 40, with higher scores indicating more severe impairment. In this study, the scale demonstrated good internal consistency (Cronbach’s α = 0.81).

#### 2.5.5. Open-Ended Questions

Participants with a history of suicide attempt(s) were asked an open-ended question regarding their perception of how their history of suicide attempt(s) impacted their experiences during the pandemic (“In what ways has surviving a suicide attempt made you more/less resilient during the pandemic?”).

### 2.6. Data Analysis

Basic descriptive statistics were calculated to characterize the demographics of the participants, stratified by the participants’ history of suicide attempt(s). In addition, independent sample *t*-tests for continuous variables and Chi-2 tests of independence for nominal and categorical variables were conducted to test the association between a history of suicide attempt(s) and participants’ sociodemographic characteristics.

The associations between a history of suicide attempts and visits with any mental health professional, psychiatric hospitalizations, and contact with hotlines were evaluated via a hurdle model, a form of count regression appropriate for infrequent outcomes [30]. This approach accounts for the high proportion of zeroes often observed in such data, modeling both the likelihood of any contact occurring and the frequency of contacts among those with at least one, providing a more accurate representation of these patterns. The association between a history of suicide attempts and pandemic impact variables was evaluated via five sets of hierarchical multiple regression. Both age and gender were included as planned covariates in all analyses, as they have been found to be associated with suicide-related outcomes [31]. All analyses were set to α at 0.05.

In this study, only data from participants who responded to the question about lifetime suicide attempts were included in the analysis. Among those who answered this question, missing data were minimal (less than 1%) and were handled via listwise deletion.

Open-ended questions were analyzed using a content-analysis approach to identify prominent themes in responses [32]. The research team engaged in multiple rounds of team discussions to operationalize variables and finalize a codebook. The codes were defined with typical and atypical exemplars to aid in determining the exclusion content for each code. Discussions and individual interpretations were informed by the multidisciplinary backgrounds of the research team (i.e., social work, behavioral science, and psychology).

## 3. Results

### 3.1. Participants

Approximately 12% (n = 159) of the participants reported having made at least one lifetime suicide attempt with the intent to die. As reported in Table 1, a greater percentage of participants with a history of suicide attempt(s) self-identified as cisgender women (62.3%), non-binary/gender non-conforming (5.7%), or bisexual/pansexual (22.6%) as compared to those without a history of suicide attempt(s). Fewer participants with a history of suicide attempt(s) reported ever being married (27.7%).

Detailed information concerning the participants’ visits to a mental health professional, psychiatric hospitalization, contacts to hotlines, and pandemic impact since March 2020 are reported in Table 2. Since the beginning of the pandemic in March 2020, 465 (31.7%) participants reported having visited a mental health professional with 2.3% (n = 34) of the participants reporting having sought psychiatric hospitalization/treatment. Nearly 10% (n = 142) of the participants reported having contacted a hotline service.

### 3.2. Group Comparisons

Using the hurdle model analytic procedure (Table 3), we tested the association between a history of suicide attempt(s) (presence/absence) and the frequency of mental health professional visits, psychiatric hospitalizations, and contact with crisis hotlines. Preliminary analyses were conducted to ensure the data were positively skewed and fit the Poisson distribution and, as such, to ensure this model fits.

Age was negatively associated with the likelihood of mental health professional visits (OR = 0.98, *p* < 0.001) and hotline contact (OR = 0.96, *p* < 0.001), indicating that younger individuals were more likely to engage with these services. Gender significantly predicted the likelihood of mental health professional visits (OR = 1.41, *p* < 0.001) and hotline contact (OR = 1.34, *p* = 0.017), with women more likely to utilize these services. However, gender showed no significant effects on psychiatric hospitalization or counts of service use across outcomes.

#### 3.2.1. Mental Health Professional Visit

A history of suicide attempts was a significant predictor of the odds of reporting any mental health professional visit during the COVID-19 pandemic (logistic regression proportion, OR = 3.14, *p* < 0.001), while controlling for age (OR = 0.98, *p* < 0.001) and sex/gender (OR = 1.41, *p* < 0.001). Thus, a history of suicide attempt(s) was associated with 3.14 times higher odds of having utilized a mental health professional. A history of suicide attempts also predicted greater counts of visits to a mental health professional (counts portion, OR = 1.66, *p* < 0.001), while controlling for age (OR = 1.00, *p* = 0.69) and gender (OR = 1.14, *p* = 0.12). This indicated that there was a 66% increase in the number of mental health professional visits for those with a history of suicide attempts.

#### 3.2.2. Psychiatric Hospitalization

A history of suicide attempts was a significant predictor of the odds of reporting any psychiatric hospitalization during the COVID-19 pandemic (logistic regression portion, OR = 7.13, *p* < 0.001), above and beyond age (OR = 0.98, *p* = 0.24) and sex/gender (OR = 0.98, *p* = 0.95). Thus, a history of suicide attempt(s) was associated with 7.13 times higher odds of having been hospitalized for psychiatric reasons during the COVID-19 pandemic. However, a history of suicide attempts did not predict greater counts of hospitalizations (counts portion, OR = 0.87, *p* = 0.86).

#### 3.2.3. Contact with Hotlines

A history of suicide attempts was a significant predictor of the odds of reporting any contact with hotlines during the COVID-19 pandemic (logistic regression portion, OR = 3.54, *p* < 0.001), above and beyond age (OR = 0.96, *p* < 0.001) and sex/gender (OR = 1.34, *p* = 0.02). Thus, a history of suicide attempt(s) was associated with 3.54 times higher odds of having contacted any type of hotline during the COVID-19 pandemic. A history of suicide attempts did not predict greater counts of those crisis hotline contacts (counts portion, OR = 1.58, *p* = 0.31).

### 3.3. Pandemic Impact

Hierarchical multiple regression was used to determine whether a history of suicide attempts was associated with pandemic impact variables (Table 4). Preliminary analyses were conducted to ensure no violation of the assumptions of normality, linearity, multicollinearity, and homoscedasticity. Age and gender were entered in Step 1 for each model as control variables, and a history of suicide attempts was entered in Step 2.

Across all models, age emerged as a significant and consistent predictor of pandemic-related effects (*p* < 0.001). Specifically, age was negatively associated with the COVID-19 impact on functioning, despair, and impairment variables and was positively associated with optimism about overcoming the pandemic. In this study, younger individuals reported greater negative pandemic effects, including higher levels of COVID-19 impact on functioning, feelings of despair, and impairment, while older individuals reported less negative effects and greater optimism about overcoming the pandemic.

Gender demonstrated significant effects in Step 1 for the COVID-19 impact on functioning and optimism about overcoming the pandemic. However, in Step 2, when a history of suicide attempts was included, gender remained significant only for the COVID-19 impact on functioning and became nonsignificant for optimism. In this study, men reported greater optimism about overcoming the pandemic and lower levels of COVID-19 impact on functioning compared to women.

#### 3.3.1. COVID-19 Impact on Functioning

The total variance explained by the model as a whole was 4.5%, F(3, 1293) = 20.17, *p* < 0.001. The history of suicide attempts explained an additional 2.0% of the variance in the COVID-19 impact on functioning variable, after controlling for age and gender, ΔR^2^ = 0.02, F change (1, 1293) = 26.78, *p* < 0.001.

#### 3.3.2. Optimism About Overcoming Pandemic

The total variance explained by the model as a whole was 3.2%, F(3, 1325) = 14.76, *p* < 0.001. A history of suicide attempts explained an additional 0.9% of the variance in optimism about overcoming pandemic variable, after controlling for age and gender, ΔR^2^ = 0.009, F change (1, 1325) = 12.40, *p* < 0.001.

#### 3.3.3. Feelings of Despair Due to Pandemic

The total variance explained by the model as a whole was 5.1%, F(3, 1304) = 23.26, *p* < 0.001. A history of suicide attempts explained an additional 2.3% of the variance in feelings of despair due to pandemic variable, after controlling for age and gender, ΔR^2^ = 0.02, F change (1, 1304) = 31.75, *p* < 0.001.

#### 3.3.4. Impairment Due to Pandemic

The total variance explained by the model as a whole was 4.9%, F(3, 1315) = 22.42, *p* < 0.001. A history of suicide attempts explained an additional 2.0% of the variance in impairment due to pandemic variable, after controlling for age and gender, ΔR^2^ = 0.02, F change (1, 1315) = 27.53, *p* < 0.001.

### 3.4. Resilience and Open-Ended Questions

Individuals who endorsed a history of suicide attempt(s) reported that they believed that having survived a suicide attempt definitely (26.5%) or probably (31.1%) made them more resilient, 21.9% of individuals reported that it might or might not have made them more resilient, 13.9% reported probably not, and 6.6% reported definitely not being more resilient. When asked to comment on their resilience, several themes emerged in the 156 participants who responded to this query.

The most common theme related to resilience was that surviving a suicide attempt made the individual see things differently and changed their perspective on life. This is exemplified in a statement such as 

It made me realize that while things may seem bad and things may be really bad at the time, things can and will get better. It’s made me more optimistic. I think it’s hardened me a bit, but in a good way. I may get figuratively knocked down but I’ll keep getting back up. And I think the other thing it changed was accepting and acknowledging when I am struggling with a problem or emotion, etc. It’s made me more self-aware which has then in turn allowed me to do experience more things in life.

Another common theme was that the suicide attempt taught the individual that they can handle more pain and suffering and reminded them they have been through worse. Examples of this theme include a reflection “Maybe it’s made me more resilient, because I know I survived wanting not to be around and I fight every day to stay.” Another participant wrote, “Just knowing that I’ll never feel as bad as I did when I was suicidal, gives me hope that I can handle the current situation, knowing that it isn’t my rock bottom mentally.”

Relatedly, some participants wrote about how they felt optimistic and found a silver lining in the theme of surviving a suicide attempt made the individual have more insight into their mental health and what they need. One participant reported, “I don’t think it has made me more resilient but it made me seek help which was what I definitely needed.” Another participant wrote

When you get over that much pain I think it could make you stronger. One thing that has happened is that I am now investigating my thoughts, feelings and actions. I’m also in therapy and back on my medication. I don’t want to ever get that bad again so this time I am doing everything I can to get better.

However, for others who survived a suicide attempt, the pandemic seemed to worsen their outlook on life. This sentiment was expressed as one participant wrote, “I think it has made me less resilient because when times get hard I always think about the possibility of doing it now.” Another person wrote, “I think hardship and pain like I’ve experienced has only ever made me worse. Adversity doesn’t make me stronger, it just hurts me, and that hurt doesn’t always heal right. I think I’d be more resilient if I had less damage in the past.” For some, the idea of suicide was still very much present “I don’t think it has made me more resilient. I think suicide is just one tool in my toolkit. It is always an option on the table, in the back of my mind. Regardless of the pandemic, there will always be hard times and when things get too hard, I can always commit [sic] suicide.” In fact, a few participants took comfort in knowing that suicide is always an option for them. “Like I said before, now I know that if everything goes wrong I can just kill myself and be done with it all. I don’t worry as much because I know that is an option.”

## 4. Discussion

This is the first study to have assessed different outcomes in coping and mental health services utilization between individuals with a history of suicide attempt(s) and those without this history during the COVID-19 pandemic. Consistent with prior research, nearly 12% of participants in the study reported having made at least one suicide attempt in their lifetime. A sizable portion of all respondents (32%) reported utilizing services from a mental health professional, with fewer reporting crisis-hotline service (9.7%) and psychiatric hospitalization utilization (2.3%).

Since the start of the COVID-19 pandemic in March 2020, participants with a history of suicide attempt(s) were more likely to seek services from mental health professionals, psychiatric hospitals, and crisis hotlines. Specifically, our findings showed that individuals with a history of suicide attempt(s) were not only approximately three times more likely to see a mental health professional (virtually or in person) but also made 66% more visits than those without a history of suicide attempt(s). These results reflect existing theories suggesting that individuals with a history of suicide attempt(s) may experience heightened distress and an increased need for professional support, particularly during periods of societal upheaval. Suicide-attempt survivors were also seven times more likely to have been hospitalized for psychiatric reasons during the pandemic; however, only about nine percent of suicide-attempt survivors were hospitalized. These patterns align with past research indicating that suicide-attempt survivors often have more severe mental health symptoms and are more likely to engage with acute care settings.

Additionally, although three and a half times more likely to make any contact with the crisis hotlines, participants with a history of suicide attempt(s) did not make a significantly higher number of those contacts than those without a history of suicide attempt(s). This discrepancy raises important questions about barriers to repeated hotline use, such as satisfaction with initial interactions or stigma. Future research should explore whether the increased service use observed in this study reflects greater distress, enhanced willingness to seek help, or a combination of both factors.

This study also examined variables related to how individuals coped during the pandemic. Participants with a history of suicide attempt(s) reported worse daily functioning, more despair and impairment, and less optimism during the pandemic. Although the history of suicide attempt(s) predicted higher distress, it explained very little of the unique variance in coping. Individuals’ histories of suicide attempts did predict a more negative impact of the pandemic on those individuals, but it indicated low explanatory power, and while statistically significant, these results may not be clinically meaningful. On the other hand, in the face of the pandemic, almost 60% of suicide-attempt survivors reported that having survived a suicide attempt had made them more resilient. Thus, certain individuals who lived through a suicide attempt might have been better equipped and readily available to seek help and endure additional distress during the pandemic.

Our qualitative analyses revealed that most participants with a history of suicide attempt(s) attribute their resilience to changing their perspective, handling more pain and suffering, and gaining more insight into their mental health after surviving a suicide attempt. However, some suicide-attempt survivors disclosed that having made a suicide attempt made their outlook on life worse and they find comfort in knowing that suicide is always an option for them.

The findings of this study indicate that people with lived experiences of suicide might have struggled more during the first year of the pandemic. They reported worse overall functioning during the pandemic, including more visits to mental health services and psychiatric hospitalizations, and less optimism, underscoring the lack of hope in this population. Consequently, public health officials should seek strategies to monitor and support those with histories of suicide attempt(s) during global pandemics and other long-term natural disasters because they might be experiencing more distress and might need more support and mental health services than others during such times. Future research should also investigate the long-term implications of increased mental health service use during the pandemic among suicide-attempt survivors. Additionally, there is a need for studies that incorporate objective behavioral measures and explore state-level policy effects on mental health outcomes during crises. Addressing these gaps will deepen our understanding of how to support vulnerable populations during periods of widespread adversity.

There are several important limitations in the present study. The nature of convenience sampling methods means that the present results cannot be generalized to the wider population. The online recruitment strategy through CloudResearch Approved List may have also increased self-selection bias and made it more difficult for individuals with limited technology access, impacting the results’ generalizability. The data also relied solely on self-report with no behavioral measures or standard measures of mental health symptoms or history of functioning prior to the pandemic available. This presents the possibility of misremembering events, feelings, and experiences—particularly for retrieving memories during a period of high stress from the pandemic. Furthermore, the pandemic impacted people differently based on the state of residence, as some states had different levels of restrictions on daily living (e.g., restaurant closures, mask mandates). These differences in policy would impact the degree to which individuals’ lives were altered and result in different levels and types of stress.

## 5. Conclusions

This study highlights the increased distress and service utilization among individuals with a history of suicide attempt(s) during the first year of the COVID-19 pandemic. These individuals were more likely to engage with mental health professionals, experience psychiatric hospitalization, and contact crisis hotlines, underscoring their increased vulnerability during large-scale public health emergencies. While many survivors reported worse functioning and lower optimism, nearly 60% endorsed that surviving a suicide attempt had contributed to their personal resilience. Their self-reported resilience suggests opportunities for strength-based interventions. These findings underscore the importance of tailoring mental health policies and practices to address the unique needs of this vulnerable population during times of global crisis.

Qualitative questions offered additional insights, emphasizing that many survivors reported increased understanding, shifts in perspective, and emotional resilience after their attempts. However, some participants expressed ongoing existential ambivalence, pointing out the lingering presence of suicidal thoughts or a sense of reassurance in knowing that suicide remains an option. The results underscore an urgent need for public health and mental health systems to proactively support individuals with lived experience of suicide, particularly during prolonged crises such as global pandemics. Clinicians, researchers, and policymakers must consider the unique service needs, coping challenges, and resilience mechanisms of this group when developing crisis response frameworks.

Future research should examine the long-term effects of increased service utilization during the pandemic and assess whether this reflects elevated distress, increased help-seeking behaviors, or both. Additionally, studies should incorporate objective behavioral data and consider regional policy variations that may influence individual experiences. Addressing these gaps will be critical in designing responsive systems of care that better serve those most at risk during future periods of societal disruption.

## Figures and Tables

**Table 1 ijerph-22-01072-t001:** Sociodemographic characteristics.

	Hx of SAs (*n* = 159)	No Hx of SAs (*n* = 1192)	
	** *n* ** **(%)**	** *n* ** **(%)**	** *p* ** ** ^f^ **
Age in years, *M(SD)*	38.55 (10.6)	40.47 (13.1)	0.076
Gender ^a^			
Cisgender woman	99 (62.3)	613 (51.4)	
Cisgender man	44 (27.7)	538 (45.1)	
Non-binary/gender non-conforming	9 (5.7)	20 (1.7)	
Transgender man	1 (0.6)	3 (0.3)	
Transgender woman	3 (1.9)	2 (0.2)	
More than one gender	0	3 (0.3)	
Not listed	1 (0.6)	0	<0.001 ^g^
Race ^b^			
White (Caucasian)	126 (79.2)	920 (77.2)	
Asian, Asian-American, or Asian-Canadian	9 (5.7)	92 (7.7)	
Black (African, African-American, African-Canadian)	5 (3.1)	77 (6.5)	
Mixed	11 (6.9)	36 (3.0)	
Native American, Alaskan Native, Inuit, First Nations, and/or Métis	1 (0.6)	5 (0.4)	
Middle Eastern and/or North African	1 (0.6)	5 (0.4)	
Native Hawaiian and/or Pacific Islander	0	1 (0.1)	
Not listed	0	1 (0.1)	0.083 ^h^
Latinx and/or Hispanic	5 (3.1)	79 (6.6)	0.088
Sexual orientation ^c^			
Heterosexual/straight	106 (66.7)	1031 (86.5)	
Homosexual/gay/lesbian	12 (7.5)	44 (3.7)	
Bisexual/pansexual	36 (22.6)	96 (8.1)	
Asexual	4 (2.5)	14 (1.2)	
Not listed	1 (0.6)	1 (0.1)	<0.001
Marital status ^d^			
Single	61 (38.4)	455 (38.2)	
Partnered	34 (21.4)	149 (12.5)	
Married	44 (27.7)	473 (39.7)	
Divorced	13 (8.2)	80 (6.7)	
Widowed	2 (1.3)	16 (1.3)	
Separated	4 (2.5)	11 (0.9)	0.004
Education ^e^			
Some high school	1 (0.6)	5 (0.4)	
High school	60 (37.7)	331 (27.8)	
Bachelor’s Degree	70 (44.0)	571 (47.9)	
Master’s Degree	17 (10.7)	168 (14.1)	
Professional School (MD, JD, MBA)	5 (3.1)	37 (3.1)	
Trade school	3 (1.9)	59 (4.9)	
Doctorate	0	14 (1.2)	0.076 ^i^

*Note*. Hx of SAs = history of suicide attempts. Eighty-four participants preferred not to answer the question about the history of suicide attempts. Thirty-two participants did not progress through the survey far enough to answer the question about the history of suicide attempts. ^a^ Eighteen participants preferred not to answer this question and one participant did not progress through the survey far enough to answer this question. ^b^ Eight participants preferred not to answer this question and sixty-two participants skipped this question. ^c^ Eleven participants preferred not to answer this question. ^d^ Fifteen participants preferred not to answer this question. ^e^ Sixteen participants preferred not to answer this question. ^f^ Participant characteristics were compared via independent sample *t* tests for continuous variables, and Χ^2^ tests of independence. ^g^ More than one gender and not listed genders were combined into a single category given the counts of zero for these categories among participants in both groups. ^h^ NHOPI and not listed in the No Hx of SAs category were excluded from analysis given the absence of any participants with Hx of SAs in this category. ^i^ Doctorate was combined with Professional School in the No Hx of SAs category given the counts of zero for this category among participants with Hx of SAs.

**Table 2 ijerph-22-01072-t002:** MH Visits, hospitalizations, hotlines and pandemic impact: overall and by history of suicide attempts.

	Hx of SAs	No Hx of SAs	Overall
	***n*** **(%)**	***n*** **(%)**	***N*** **(%)**
Any mental health professional visit (via telehealth or in person)	94 (59.1)	337 (28.3)	465 (31.7)
Psychiatric hospitalization	15 (9.4)	17 (1.4)	34 (2.3)
Contact with hotlines	39 (24.5)	92 (7.7)	142 (9.7)
	*M (SD)*	*M (SD)*	*M (SD)*
If any,			
Number of visits to any mental health professional	20.5 (22.4)	13.0 (16.0)	15.1 (18.5)
Number of hospitalizations	1.5 (1.1)	1.6 (1.2)	1.5 (1.1)
Number of contacts to hotlines	7.7 (19.6)	5.4 (17.0)	5.8 (17.3)
Overall impact of COVID-19 on functioning	37.8 (7.8)	33.9 (7.7)	34.4 (7.8)
Optimism about overcoming pandemic	21.3 (5.2)	22.9 (4.3)	22.7 (4.5)
Feelings of despair due to pandemic	9.5 (3.7)	7.8 (3.1)	8.0 (3.3)
Impairment due to pandemic	14.6 (6.2)	11.8 (5.6)	12.1 (5.7)

*Note*. MH = Mental health. Hx of SAs = History of suicide attempts.

**Table 3 ijerph-22-01072-t003:** Hurdle model analyses.

	Logistic Regression Portion			Counts Portion		
	ß (*SE*)	*OR* (95% CI)	*p*	ß (*SE*)	*RR* [95% CI]	*p*
Dependent variable: Mental-health professional visit (via telehealth or in person)						
Age	−0.02 (0.01)	0.98 [0.97, 0.99]	<0.001	−0.00 (0.01)	1.00 [0.99, 1.01]	0.694
Gender	0.34 (0.10)	1.41 [1.16, 1.70]	<0.001	0.13 (0.08)	1.14 [0.97, 1.35]	0.118
Hx of SAs	1.14 (0.18)	3.14 [2.21, 4.46]	<0.001	0.51 (0.15)	1.66 [1.24, 2.22]	<0.001
Dependent variable: Psychiatric hospitalization						
Age	−0.02 (0.02)	0.98 [0.95, 1.01]	0.242	0.03 (0.04)	1.03 [0.96, 1.11]	0.409
Gender	−0.17 (0.25)	0.98 [0.60, 1.62]	0.947	0.48 (0.79)	1.62 [0.35, 7.62]	0.539
Hx of SAs	1.96 (0.37)	7.13 [3.43, 14.84]	<0.001	−0.14 (0.81)	0.87 [0.18, 4.23]	0.861
Dependent variable: Contact with hotlines						
Age	−0.04 (0.01)	0.96 [0.94, 0.98]	<0.001	0.01 (0.02)	1.01 [0.97, 1.06]	0.535
Gender	0.29 (0.12)	1.34 [1.01, 1.71]	0.017	−0.81 (0.21)	0.44 [0.29, 0.67]	<0.001
Hx of SAs	1.26 (0.23)	3.54 [2.25, 5.56]	<0.001	0.46 (0.46)	1.58 [0.65, 3.84]	0.314

*Note*. Hx of SAs = history of suicide attempts. CI = confidence interval; *SE* = standard error or unstandardized ß coefficients; *OR* = odds ratio; *RR* = rate ratio.

**Table 4 ijerph-22-01072-t004:** Hierarchical multiple regression of history of suicide attempts on pandemic effects.

	B (*SE)*	ß	*p*	Adj. R^2^	R^2^	ΔR^2^
Dependent variable: COVID-19 impact on functioning						
Step 1				0.023	0.025	0.025 **
Age	−0.07 (0.02)	−0.11	<0.001			
Gender	1.38 (0.33)	0.12	<0.001			
Step 2				0.042	0.045	0.020 **
Age	−0.06 (0.02)	−0.10	<0.001			
Gender	1.51 (0.33)	0.10	<0.001			
Hx of SAs	3.41 (0.66)	0.14	<0.001			
Dependent variable: Optimism about overcoming pandemic						
Step 1				0.022	0.023	0.023 **
Age	0.04 (0.01)	0.12	<0.001			
Gender	−0.63 (0.19)	−0.09	<0.001			
Step 2				0.030	0.032	0.009 **
Age	0.04 (0.01)	0.12	<0.001			
Gender	−0.54 (0.19)	−0.08	0.004			
Hx of SAs	−1.33 (0.38)	−0.10	<0.001			
Dependent variable: Feelings of despair due to pandemic						
Step 1				0.026	0.028	0.028 **
Age	−0.04 (0.01)	−0.15	<0.001			
Gender	0.38 (0.14)	0.08	0.005			
Step 2				0.049	0.051	0.023 **
Age	−0.04 (0.01)	−0.14	<0.001			
Gender	0.26 (0.14)	0.05	0.053			
Hx of SAs	0.1.55 (0.28)	0.15	<0.001			
Dependent variable: Impairment due to pandemic						
Step 1				0.027	0.029	0.029 **
Age	−0.07 (0.01)	−0.16	<0.001			
Gender	0.47 (0.24)	0.05	0.051			
Step 2				0.046	0.049	0.020 **
Age	−0.07 (0.01)	−0.15	<0.001			
Gender	0.30 (0.24)	0.03	0.209			
Hx of SAs	2.54 (0.48)	0.14	<0.001			

*Note*. Hx of SAs = History of suicide attempts. * *p* < 0.05; ** *p* < 0.01.

## Data Availability

The participants of this study did not give written consent for their data to be shared publicly, so due to the sensitive nature of the research, supporting data is not available.

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
