# Peer review of "Examining the Impact of the COVID-19 Pandemic on Suicide-Attempt Survivors"

_ijerph, 2025, doi:10.3390/ijerph22071072_

Round 1
Reviewer 1 Report
Comments and Suggestions for Authors
REVIEW REPORT
The research proposal is interesting, as it contributes to disseminating information about the COVID stage and what happened during and after it.
However, there are some observations that we will mention:
Regarding the introduction, it is requested to improve the theoretical foundations that help contextualize the general overview and current situation of what is investigated. Furthermore, ensure that the main objective of the investigation is declared in a standardized manner and the same statement is always used.
Confusing methodology: The construction of the manuscript in the methodology section requires a substantial improvement, which includes the type and design of the research carried out, the declaration of population, sample and sampling used, the validity and reliability of the instruments used, since they declare that they have been modified or served as 'inspiration' for other instruments, which generates difficulty in understanding which instruments were finally used. In addition, constructs such as resilience are mentioned that have not been evaluated through psychological instruments, it is only deduced through a question and that then serves to build the conclusion. On the other hand, carefully declare the procedure carried out, which allows us to form an idea of ​​how the work was carried out. Finally, include the ethical aspects, which declare the review report of the ethics committee that reviewed and approved this research.
In lines 94-96 it is stated that previous studies demonstrate the reliability of the MTURK platform for collecting high-quality psychological and behavioral research data, based on three references from the years 2010, 2011 and 2013, that is, from 14 years ago on average. Upon consulting them, it is observed that the three references declare doubts about the reliability and validity of the information provided by participants recruited in online labor markets. In this sense, it is suggested to update references from the last 5 years that support the use of online labor markets, in order to respond to surveys that serve scientific research.
Added to that, we have a concern; If the participant receives payment for answering the instruments of the research in question, is informed consent justified? It is suggested that arguments be sought to support the use of consent in these cases.
Regarding the conclusion, it must correspond to a general construction that is consistent with the title, responds to the objectives and includes an argumentative statement from the authors. In addition, the limitations found during the investigation must be included,
The references presented are 32. Of which only 8 (25%) correspond to database journals indexed in the last 5 years. It is suggested to substantially increase the references of indexed data journals to a minimum of 40%. Additionally, review the drafting proposal for all references to unify its style.
Finally, a complete review of the manuscript is recommended, as well as a reconstruction of it based on the suggestions given so that the work can be saved and aspire to publication.
Author Response
The research proposal is interesting, as it contributes to disseminating information about the COVID stage and what happened during and after it.
However, there are some observations that we will mention:
Comments 1: Regarding the introduction, it is requested to improve the theoretical foundations that help contextualize the general overview and current situation of what is investigated. Furthermore, ensure that the main objective of the investigation is declared in a standardized manner and the same statement is always used.
Response 1: Thank you for your feedback. We have carefully reviewed the introduction to improve clarity and ensure that the general overview is well contextualized within the current theoretical foundations. Additionally, we have made sure that the main objective of the investigation is clearly stated in a standardized and consistent manner throughout the manuscript.
Comments 2: Confusing methodology: The construction of the manuscript in the methodology section requires a substantial improvement, which includes the type and design of the research carried out, the declaration of population, sample and sampling used, the validity and reliability of the instruments used, since they declare that they have been modified or served as 'inspiration' for other instruments, which generates difficulty in understanding which instruments were finally used. In addition, constructs such as resilience are mentioned that have not been evaluated through psychological instruments, it is only deduced through a question and that then serves to build the conclusion. On the other hand, carefully declare the procedure carried out, which allows us to form an idea of how the work was carried out. Finally, include the ethical aspects, which declare the review report of the ethics committee that reviewed and approved this research.
Response 2: Thank you for this thorough and constructive feedback. We agree that greater clarity was needed in the methodology section, and we have revised it to describe the study's design as a cross-sectional survey explicitly. We have clarified the population, sampling approach, and sample characteristics and provided additional details on the instruments used. For relevant measures, we have included information on their reliability. We also clarified when items were adapted or inspired by previous tools to distinguish them from validated instruments. Regarding the resilience construct, we acknowledge that it was not measured using a standardized psychological scale and have provided a more detailed description of the single-item approach used to assess perceived resilience. Additionally, we have expanded our description of the study procedures to ensure the methodology is understandable and transparent, and we included ethical considerations and Institutional Review Board (IRB) approval information to address this concern.
Comments 3: In lines 94-96 it is stated that previous studies demonstrate the reliability of the MTURK platform for collecting high-quality psychological and behavioral research data, based on three references from the years 2010, 2011 and 2013, that is, from 14 years ago on average. Upon consulting them, it is observed that the three references declare doubts about the reliability and validity of the information provided by participants recruited in online labor markets. In this sense, it is suggested to update references from the last 5 years that support the use of online labor markets, in order to respond to surveys that serve scientific research.
Response 3: Thank you for this fair and insightful comment. We agree that the citations originally included to support the reliability of MTurk data were outdated. In response, we have updated the references to include more recent literature from more recent years that better reflects current evaluations of data quality and participant reliability on online labor markets. These newer sources provide stronger and more current support for the use of platforms like MTurk and CloudResearch in behavioral and psychological research.
Comments 4: Added to that, we have a concern; If the participant receives payment for answering the instruments of the research in question, is informed consent justified? It is suggested that arguments be sought to support the use of consent in these cases.
Response 4: Thank you for your comment. You are correct to raise this important consideration. Participants were indeed consented to this survey, and this process is described in the manuscript. While participants received compensation for their time, informed consent was obtained in accordance with ethical research standards, clearly stating the voluntary nature of participation and the right to withdraw at any time without penalty. We have ensured that this is clearly articulated in the manuscript to address any concerns about the legitimacy of consent in compensated online research.
Comments 5: Regarding the conclusion, it must correspond to a general construction that is consistent with the title, responds to the objectives and includes an argumentative statement from the authors. In addition, the limitations found during the investigation must be included,
Response 5: Thank you for this constructive feedback. We agree that the original conclusion needed improvement, and we have expanded it to better align with the study’s title and objectives. The revised conclusion now includes a more comprehensive synthesis of the study’s findings.
Comments 6: The references presented are 32. Of which only 8 (25%) correspond to database journals indexed in the last 5 years. It is suggested to substantially increase the references of indexed data journals to a minimum of 40%. Additionally, review the drafting proposal for all references to unify its style.
Response 6: Thank you for your feedback. We appreciate your attention to the recency and consistency of our references. In response, we have reviewed and updated the reference list. We also revised the formatting of all references to ensure consistency and adherence to the appropriate citation style throughout the manuscript.
Comments 7: Finally, a complete review of the manuscript is recommended, as well as a reconstruction of it based on the suggestions given so that the work can be saved and aspire to publication.
Response 7: Thank you for this constructive feedback. We took your recommendation seriously and have completed a thorough review and revision of the manuscript, addressing each of the suggestions provided. We appreciate your guidance in helping us strengthen the clarity, structure, and overall quality of the paper, and we hope the revised version reflects these improvements and is now suitable for publication.
Reviewer 2 Report
Comments and Suggestions for Authors
This article provides an exceptional methodological approach to a sensitive and integral area of study. It was a privilege to review this unique study that is an undeniable benefit to all fields addressing suicide. A few minor revisions would help to refine this manuscript.
Abstract
The abstract is outlined in clear, understandable language and identifiable points of interest. It would benefit the audience to have the study’s location (United States) more precisely stated rather than inferring by saying “50 states.”
It would also be helpful to clarify the latter part of the outcomes sentence, “while 31.1% probably felt the same.” Does this mean these individuals believed that surviving their previous suicide attempt made no perceived difference in their resiliency?
And who are “These participants” in the last sentence? This seems to refer to the “31.1%” group. Does this actually refer to all participants or a specific group? I recommend clarifying this reference.
Introduction
The introduction was clearly outlined and well-cited. However, please use quotation marks (“ ”) for any direct quotes, as seen in Lines 64–68, beginning, “This is not a drill…” I especially appreciated the “Current Study” identification marking the start of the last paragraph.
Materials and Methods
The thoroughness of the methodology narrative is exceptional. No revisions needed.
Results
The results are very well detailed and are unquestionable contributions to the field. The sociodemographic table with comparative suicide attempt and no suicide attempt histories alone is imperative to furthering knowledge on this subject. The entirety of the results analyzed offer a unique analysis of a complex and relatively understudied subject that can be applied to extreme social events similar to pandemic circumstances. No revisions needed.
Discussion
The discussion section narrates clear explanations of the study’s multifaceted results with key supporting evidence from existing literature. The explanations of key findings were very well outlined, as with the implications for future research and limitations. No revisions needed.
Conclusions
The conclusion is succinct and includes some key aspects of this study (objective, findings). However, it would be beneficial to include a couple more sentences on the study’s background (why is it important that the setting was in the United States?) and implications for future studies or policies related to suicidality and mental well-being.
Author Response
Comments 1: This article provides an exceptional methodological approach to a sensitive and integral area of study. It was a privilege to review this unique study that is an undeniable benefit to all fields addressing suicide. A few minor revisions would help to refine this manuscript.
Response 1: We are sincerely grateful for the reviewer’s kind words and thoughtful review. We greatly appreciate the recognition of the methodological approach and the importance of this work. It is incredibly encouraging to hear that you view the study as a meaningful contribution to the field of suicide research. Thank you for your helpful suggestions—we have addressed the minor revisions you recommended to refine the manuscript further.
Abstract
Comments 2: The abstract is outlined in clear, understandable language and identifiable points of interest. It would benefit the audience to have the study’s location (United States) more precisely stated rather than inferring by saying “50 states.”
Response 2: We agree with the reviewer’s suggestion and have revised the abstract to explicitly state that the study was conducted in the United States, rather than referring to the “50 states” (p. 1):
An online nationwide sample of 1,351 adults from the United States completed questionnaires from May 26 to June 25, 2021.
Comments 3: It would also be helpful to clarify the latter part of the outcomes sentence, “while 31.1% probably felt the same.” Does this mean these individuals believed that surviving their previous suicide attempt made no perceived difference in their resiliency?
Response 3: Thank you for catching this and pointing out the lack of clarity. You are correct in your interpretation—the individuals in this group reported no perceived difference in their resilience after surviving a suicide attempt. We have revised the sentence to clarify this point in the manuscript (p. 1):
Notably, 57.6% of these individuals believed surviving a suicide attempt made them more resilient, while 21.9% expressed uncertainty about its impact on their resilience.
Comments 4: And who are “These participants” in the last sentence? This seems to refer to the “31.1%” group. Does this actually refer to all participants or a specific group? I recommend clarifying this reference.
Response 4: We agree that the phrasing lacked clarity, and we appreciate the reviewer highlighting this point. We have revised the sentence to clearly specify the group being referenced, ensuring that it is now unambiguous and easier to follow (p. 1):
In sum, participants with a history of suicide attempt(s) reported more depressive symptoms, worse daily functioning, more despair, less optimism, and greater service utilization during the pandemic, yet many also cited increased resilience due to their suicide history.
Introduction
Comments 5: The introduction was clearly outlined and well-cited. However, please use quotation marks (“ ”) for any direct quotes, as seen in Lines 64–68, beginning, “This is not a drill…” I especially appreciated the “Current Study” identification marking the start of the last paragraph.
Response 5: Thank you for your positive feedback on the introduction and for pointing out the formatting issue. We agree that quotation marks should be used for direct quotes and have revised the manuscript accordingly by adding quotation marks to the quoted material in Lines 64–68 (p. 2):
For example, a popular press article authored by a person with bipolar disorder written early in the pandemic states, “This is not a drill, but for those of us who struggle with bipolar disorder like myself, or depression, or anxiety, this is not our first rodeo. We’re surprisingly copacetic. We’ve had emotional distress time after time, so we are more resilient than the normies a.k.a. people without mental illness. In other words, we’ve been training for this our whole lives”[16].
Materials and Methods
Comments 6: The thoroughness of the methodology narrative is exceptional. No revisions needed.
Response 6: Thank you for your kind feedback. We’re glad to hear that the methodology section was clear and thorough.
Results
Comments 7: The results are very well detailed and are unquestionable contributions to the field. The sociodemographic table with comparative suicide attempt and no suicide attempt histories alone is imperative to furthering knowledge on this subject. The entirety of the results analyzed offer a unique analysis of a complex and relatively understudied subject that can be applied to extreme social events similar to pandemic circumstances. No revisions needed.
Response 7: Thank you for your generous and encouraging feedback. We are grateful that you found the results section and sociodemographic table to be meaningful contributions.
Discussion
Comments 8: The discussion section narrates clear explanations of the study’s multifaceted results with key supporting evidence from existing literature. The explanations of key findings were very well outlined, as with the implications for future research and limitations. No revisions needed.
Response 8: Thank you for your thoughtful feedback. We’re pleased to hear that the discussion effectively conveyed the key findings and their implications, and that it aligned well with existing literature.
Conclusions
Comments 9: The conclusion is succinct and includes some key aspects of this study (objective, findings). However, it would be beneficial to include a couple more sentences on the study’s background (why is it important that the setting was in the United States?) and implications for future studies or policies related to suicidality and mental well-being.
Response 9: Thank you for this suggestion. We have expanded the conclusion to include additional context and have added several sentences outlining implications for future research and policy related to suicidality and mental well-being (p. 13):
This study highlights the increased distress and service utilization among individuals with a history of suicide attempt(s) during the first year of the COVID-19 pandemic. These individuals were more likely to engage with mental health professionals, experience psychiatric hospitalization, and contact crisis hotlines, underscoring their increased vulnerability during large-scale public health emergencies. While many survivors reported worse functioning and lower optimism, nearly 60% endorsed that surviving a suicide attempt had contributed to their personal resilience. Their self-reported resilience suggests opportunities for strength-based interventions. These findings underscore the importance of tailoring mental health policies and practices to address the unique needs of this vulnerable population during times of global crisis.
Qualitative questions offered additional insights, emphasizing that many survivors reported increased understanding, shifts in perspective, and emotional resilience after their attempts. However, some participants expressed ongoing existential ambivalence, pointing out the lingering presence of suicidal thoughts or a sense of reassurance in knowing that suicide remains an option. The results underscore an urgent need for public health and mental health systems to proactively support individuals with lived experience of suicide, particularly during prolonged crises such as global pandemics. Clinicians, researchers, and policymakers must consider the unique service needs, coping challenges, and resilience mechanisms of this group when developing crisis response frameworks.
Future research should examine the long-term effects of increased service utilization during the pandemic and assess whether this reflects elevated distress, increased help-seeking behaviors, or both. Additionally, studies should incorporate objective behavioral data and consider regional policy variations that may influence individual experiences. Addressing these gaps will be critical in designing responsive systems of care that better serve those most at risk during future periods of societal disruption.
Reviewer 3 Report
Comments and Suggestions for Authors
Very interesting article. An important problem of coping of people after suicide attempts during the period of threat/social trauma of Covid-19 was taken up. The research procedure was analyzed in detail. Analysis of results was correct. The authors correctly focused on the analysis of results related to the main research problem. However, they obtained very important additional results that could be taken up in the discussion, e.g. connections between sexual identity, race, education - and the frequency of suicide attempts. The application aspect of the research, especially the social one, could also be expanded in the conclusions.
Author Response
Comments 1: Very interesting article. An important problem of coping of people after suicide attempts during the period of threat/social trauma of Covid-19 was taken up. The research procedure was analyzed in detail. Analysis of results was correct. The authors correctly focused on the analysis of results related to the main research problem. However, they obtained very important additional results that could be taken up in the discussion, e.g. connections between sexual identity, race, education - and the frequency of suicide attempts. The application aspect of the research, especially the social one, could also be expanded in the conclusions.
Response 1: Thank you very much for your thoughtful and encouraging feedback. We truly appreciate your recognition of the importance of our research on coping after suicide attempts during the COVID-19 pandemic. Based on your helpful suggestions, we have expanded the conclusion to further highlight the application and social implications of our findings (p. 13):
This study highlights the increased distress and service utilization among individuals with a history of suicide attempt(s) during the first year of the COVID-19 pandemic. These individuals were more likely to engage with mental health professionals, experience psychiatric hospitalization, and contact crisis hotlines, underscoring their increased vulnerability during large-scale public health emergencies. While many survivors reported worse functioning and lower optimism, nearly 60% endorsed that surviving a suicide attempt had contributed to their personal resilience. Their self-reported resilience suggests opportunities for strength-based interventions. These findings underscore the importance of tailoring mental health policies and practices to address the unique needs of this vulnerable population during times of global crisis.
Qualitative questions offered additional insights, emphasizing that many survivors reported increased understanding, shifts in perspective, and emotional resilience after their attempts. However, some participants expressed ongoing existential ambivalence, pointing out the lingering presence of suicidal thoughts or a sense of reassurance in knowing that suicide remains an option. The results underscore an urgent need for public health and mental health systems to proactively support individuals with lived experience of suicide, particularly during prolonged crises such as global pandemics. Clinicians, researchers, and policymakers must consider the unique service needs, coping challenges, and resilience mechanisms of this group when developing crisis response frameworks.
Future research should examine the long-term effects of increased service utilization during the pandemic and assess whether this reflects elevated distress, increased help-seeking behaviors, or both. Additionally, studies should incorporate objective behavioral data and consider regional policy variations that may influence individual experiences. Addressing these gaps will be critical in designing responsive systems of care that better serve those most at risk during future periods of societal disruption.
Round 2
Reviewer 1 Report
Comments and Suggestions for Authors
having reviewed the manuscript and considering that, although the letter of response to the comments describes a new proposal, in the improved manuscript no major modifications are observed, except in very specific words and some work in constructing the conclusion.
In this sense, the revision of the "improved" manuscript is suggested in light of the observations made initially that are not observed as raised.
Author Response
Thank you for your thoughtful review and for the opportunity to revise our manuscript.
We want to clarify that we have made significant revisions to the manuscript based on the initial feedback. We ensured that all relevant references were included, and as this study addresses a unique and previously unexplored topic—the experiences of suicide attempt survivors during a national disaster such as the COVID-19 pandemic—we have incorporated all existing literature available on this specific subject.
In addition, we have previously extensively revised and clarified the Research Design and Methods section, providing as much detail as possible. The Results and Conclusion sections have also undergone multiple rounds of careful revision to strengthen clarity, relevance, and alignment with the study’s objectives and findings.
However, if there are still specific aspects the reviewer believes are lacking—such as particular references or methodological components—we would be very grateful for more detailed guidance. If the reviewer could kindly indicate what references they feel are missing or which parts of the methodology need further elaboration, we would be happy to address them directly.
We greatly appreciate your time and expertise and remain committed to producing a high-quality manuscript.
With sincere thanks,
Martina Fruhbauerova